# Testing the Triggering Hypothesis: Effect of Cognate Status on Code-Switching and Disfluencies

Anne Neveu [1,*], Margarethe McDonald [2,3] and Margarita Kaushanskaya [1]

1    Department of Communication Sciences and Disorders, University of Wisconsin-Madison, Madison, WI 53706-1380, USA
2    Department of Linguistics, University of Ottawa, Ottawa, ON K1N 6N5, Canada
3    School of Psychology, University of Ottawa, Ottawa, ON K1N 6N5, Canada
*    Correspondence: aneveu@wisc.edu

**Abstract:** "Triggered switching" is the theory that code-switching happens more often with words connected to both languages, such as cognates. Corpus analyses have supported this theory; however, they do not allow testing for directional causality. Here, we test the triggering hypothesis through a picture-naming task, and examine whether cognates trigger code-switches, as well as more subtle interference effects resulting in disfluencies. Forty English-Spanish bilinguals completed a picture-cued sentence production task in three conditions: English-only, Spanish-only, and mixed. Half of the pictures represented Spanish-English cognates. Unsurprisingly, participants were more likely to code-switch when asked to use both their languages compared to only their dominant or non-dominant language. However, participants were not more likely to switch languages for cognate than for non-cognate trials. Participants tended to be more fluent on cognate trials in the dominant and the non-dominant condition, and on non-cognate trials in the mixed-language condition, although these effects were not significant. These findings suggest that both language context and cognate status are important to consider when testing both overt switches and disfluencies in bilingual speech production.

**Keywords:** bilingualism; code-switching; cognates; disfluencies; triggered switching

## 1. Introduction

Code-switching is the act of naturally alternating between two languages within or across sentences for bilingual individuals. It has been the subject of extensive study in psycholinguistics, in part because it represents overt evidence of language coactivation in bilinguals (e.g., Hatzidaki et al. 2011; Fricke and Kootstra 2016; Green 2018; Kootstra et al. 2020; Sarkis and Montag 2021). Coactivation is a robust effect where a bilingual's two language are simultaneously active in comprehension and production, in turn leading the languages to influence each other (e.g., Marian and Spivey 2003a, 2003b; Kroll et al. 2006, 2015; Hoshino and Kroll 2008; Hatzidaki et al. 2011; Moon and Jiang 2012; Lijewska 2020).

The question of how code-switches emerge was asked in a seminal 1980 study, when Clyne suggested that code-switching is more likely to occur, or be triggered, in the presence of cross-linguistic overlap, such as in the presence of cognates, false friends, homophones, proper nouns or loan words. This formed the basis for Clyne's triggering hypothesis (1980): code-switching would occur either just before, or immediately after one of these trigger word. Research on language coactivation using cognates as a tool to reveal mechanisms of cross-language processing robustly suggests that coactivation is enhanced in the presence of words that are shared across lexicons (see Van Hell and Tanner 2012 for a review). It is hypothesized that as a result, trigger words would lead bilinguals to coactivate their languages and enhance the likelihood of code-switching in production.

To quantitatively examine the original triggering hypothesis, Broersma and De Bot (2006) ran corpus-based statistical analyses in a Dutch and Moroccan Arabic corpus, and

found that words immediately following but not preceding a trigger word were more likely to be code-switched. They additionally expanded the triggering hypothesis from the immediate word level to the clause level, calling it the adjusted triggering hypothesis. This hypothesis posits that words in the vicinity of a trigger word at the clause level may be more likely to be code-switched. The findings supported this hypothesis. Another corpus analysis involving related languages (Dutch and English) examined various types of code-switches (Broersma 2009). It compared the likelihood of a switch to occur directly around a trigger word, or at the clause level. Results at the word-level showed that words around a trigger word were more likely to be code-switched. There was no statistical difference in code-switching frequency depending on whether the trigger word preceded or followed the switch. At the clause level, similar results were found: clauses with a trigger word were more likely to contain a code-switch.

In a more recent study, Broersma et al. (2020) tested several hypotheses, including the following: (1) Cognates would facilitate clause-internal and clause-external code-switching; (2) Cognate-dense clauses would be more likely to be followed by a code-switch; and (3) A cognate would trigger a code-switch in the following clause. The study was based on a corpus of Welsh-English bilingual conversations. A clause-internal switch was defined as a clause that included both uniquely Welsh and English words, excluding cognates. A clause-external switch was defined as a clause without a trigger word that had a code-switch and that was directly next to a clause with a trigger word. Results showed that (1) Cognates were associated with clause-internal and clause-external code-switching; (2) More cognate-dense clauses had more clause-external code-switching. Shorter clauses additionally were associated with more clause-external code-switches; and (3) A clause with a cognate was more likely to be followed by a code-switch than a clause without a cognate.

These findings indicate that cognates and code-switches co-occur in spontaneous conversations. However, corpus data are necessarily acausal, and thus do not allow establishing directionality in the relationship seen between trigger words and code-switches. It may be, as suggested by trigger hypotheses, that a trigger word leads to the appearance of a code-switch; however, it may also be the case that a code-switch enhances the chances of using "trigger words". In the case of the latter, the label "trigger word" could be misleading, in that "trigger" implies that the effect originates from the word and leads to a switch, but this is not necessarily the case. Indeed, code-switching relies on dual-language activation, and as a result, words that are connected to both languages could then be more readily available to the bilingual speaker. The switch-to-trigger word hypothesis could also be explained by the fact that as a speaker is preparing to switch languages, they are announcing and/or easing this process for the listener by using cognates, as in audience design. Indeed, it has been shown that bilingual listeners can use linguistic cues to anticipate a code-switch (Fricke et al. 2016). The triggering hypothesis could be experimentally tested by inducing speakers to produce trigger words and by observing the extent to which code-switches are present. A few studies have done so using sentence production tasks and have yielded somewhat conflicting findings.

Kootstra et al. (2012) examined the role of cognates in priming code-switched utterances. They had participants repeat a sentence that contained a language switch mid-sentence which was directly preceded by a cognate or a non-cognate. Then participants were asked to make their own sentence that included a language switch, based on a presented picture. The researchers examined how often the participants switched in the same position of the sentence (e.g., directly after the verb) as where the switch happened in the prime sentence. Based on the literature suggesting that cognates facilitate language processing (e.g., Costa et al. 2000; Van Assche et al. 2009; Dijkstra et al. 2010), and the triggering hypothesis, which also supports a cognate facilitation effect, they hypothesized that switching in the same sentence position as the prime sentence would be facilitated by the presence of a cognate in the prime sentence and when the target picture was a cognate. Findings revealed that cognates facilitated a switch in the same position but

only for high-proficiency L2 speakers. In contrast, Bultena et al. (2015a) did not observe a cognate effect in their study, which used switching costs as an index of triggering instead of switch placement (Kootstra et al. 2012). Bultena et al. (2015a) examined whether switch costs were affected by language coactivation, using cognates preceding a code-switch in a shadowing task. Bilinguals were presented with sentences in their L1 and L2, and asked to listen to recorded speech and to subsequently repeat it as fast and accurately as possible. Switch costs in both directions were not influenced by the presence of cognates in this study. However, the difference in findings across Kootstra et al. (2012) and Bultena et al. (2015a) could be due to the different methods used to measure switching.

More recently, Kootstra et al. (2020) tested the presence and frequency of code-switching produced by Dutch (dominant language)-English bilinguals in a dialogue game. Participants had to produce a sentence to describe a picture that was either a cognate, a false friend, or a control. In addition, they would produce their sentence in one of two conditions: either after a confederate had made a deliberate code-switch, or not. Results suggested that participants were more likely to code-switch only if a confederate code-switched in the previous trial. This effect was stronger if the image they had to describe was of a cognate word compared to a false friend. Overall however, no significant main effect of cognate status was observed, suggesting that cognates were not more likely than control words to trigger code-switches.

Therefore, findings from corpus analyses and sentences production experiments do not fully align in elucidating the relationship between cognates and code-switches and the triggering hypothesis warrants further examination. Notably, it is possible that the presence of cognates in the context of sentence production could create interference on a more subtle scale that would not lead to an overt code-switch. For example, because a trigger word should lead to enhanced dual-language activation in a bilingual speaker, it may trigger disfluencies rather than overt switches. In his original conceptualization of the triggering hypothesis, Clyne (1980) suggested that trigger words lead to uncertainty and observed that "triggering can, of course, be regarded as a type of speech error". Evidence from corpus analyses suggests that "a code-switch is often preceded by a hesitation" (Clyne 2000). However, another corpus analysis found that up to 96% of the time code-switches were not accompanied by disfluencies (Poplack 1980). Disfluencies, defined as an unintentional interruption in speech, can take the form of a hesitation, a filled pause, or a retrace (e.g., false starts: beginning a sentence and correcting oneself to formulate the sentence differently). A logical extension of a triggering hypothesis would be to posit that because a cognate activates both lexicons, it would subtly destabilize the speaker attempting to stay within a single language, and would trigger disfluencies. The population in Clyne (1980, 2000) and Poplack (1980) are early bilinguals who are less likely to produce disfluencies generally due to high proficiency in both their languages. Disfluencies may be more readily detectable in speakers who are not as proficient in their L2.

In summary, the evidence for the triggering hypothesis is mixed: corpus analyses suggest that code-switches may co-occur with cognates, and experimental studies overall do not find that cognates trigger code-switching, except potentially for highly proficient L2 speakers. Disfluencies produced in the vicinity of trigger words in bilingual productions have not previously been experimentally investigated.

The present study was designed to examine both whether cognates would trigger code-switching in a cued sentence production paradigm, and whether cognates would lead to an increase in the amount of disfluencies. In the cued sentence production paradigm, bilingual participants were cued with a picture that represented either a cognate or a non-cognate word. This cueing paradigm offers one advantage over examining the natural occurrence of code-switching in corpus data in that we have control over the directionality of the effect (with the limitation of potential spill-over/priming effects between trials). However, the disadvantage of such an approach is that increasing control over the conditions of production decreases the ecological validity of the task.

Because language dominance appears to play a role in triggering effects, and in the patterns of switching (Bultena et al. 2015a, 2015b), we manipulated language context of

the task, such that all participants produced sentences in the dominant language, the non-dominant language, and in a mixed condition, where they were asked to use both of their languages to a similar extent. The purpose of the instructions in the mixed language conditions was to maximize language switching (as in Gollan et al. 2014). Our research questions for this study were as follows:

1.  Do cognates trigger code-switches?

    We hypothesized that if cognates trigger code-switches, then more switches would be observed in picture descriptions where participants were cued to produce cognates than where participants were cued to produce non-cognate (control) words.

2.  Do cognates trigger disfluencies?

    We hypothesize that if cognates disrupt fluency, then more disfluencies would be observed in picture descriptions where participants were cued to produce cognates, rather than control words.

3.  Do cognate effects vary by language of the task?

    We hypothesized that cognate effects would be strongest in the mixed condition, because the dual-language context of this condition would be most conducive to cross-linguistic activation and interference.

## 2. Materials and Methods

### 2.1. Participants

Sixty-six adults who had knowledge of both English and Spanish participated in the study. Participants were screened for English and Spanish knowledge prior to the task and those who reported home or school exposure to both languages were asked to participate. Participants completed the Language Experience and Proficiency Questionnaire (LEAP-Q; Marian et al. 2007) which probed for dominant language, self-ratings of language proficiency in each language, and acquisition of languages besides English and Spanish. Since Germanic and Romance languages were likely to have cognates with English and Spanish, participants who rated their speaking abilities in any such languages other than English and Spanish as a 5 or above on a scale of 0–10 were excluded (10 participants).

Further, some participants did not have sufficiently strong abilities in English or Spanish, making the task difficult to complete in both languages. Participants who rated their speaking abilities as below 5 in English or Spanish were excluded (7 participants). Further exclusions included technical errors leading to no sound recordings (5 participants), and labeling less than 33% of items with the correct label in both languages on test trials (3 participants). Remaining participants were classified as English or Spanish dominant based on self-report (first question on the LEAP-Q). One participant reported being equally dominant in both English and Spanish, and since they did not fit into either dominance group, they were excluded.

As a result, 40 participants (15 males, 25 females) were retained in the final analysis. This included both participants who were Spanish dominant (12 participants) and English dominant (28 participants). All participants completed the Peabody Picture Vocabulary Test, 4th Edition (PPVT-4, Dunn and Dunn 2007) to index English receptive vocabulary abilities. Spanish proficiency was indexed by the Test de Vocabulario en Imagenes Peabody (TVIP; Dunn et al. 1986). Due to participants falling outside of the normative age for the TVIP, the whole test was administered, and percent accuracy was computed. Finally, participants completed the Kaufman Brief Intelligence Test, 2nd Edition (KBIT-2) Matrices subtest (Kaufman and Kaufman 2004) as a measure of nonverbal intelligence. Scores on tests and self-proficiency ratings are available in Table 1. Scores on the vocabulary tests aligned with the language dominance classification such that the Spanish dominant group had significantly higher Spanish scores than the English-dominant group ($t(24.1) = 4.87$, $p < 0.001$), and the English dominant group had significantly higher English scores than the Spanish-dominant group ($t(13.7) = 4.13$, $p < 0.01$).

**Table 1.** Participant Characteristics: Mean (SD).

|  | Spanish Dominant | English Dominant | All Participants |
|---|---|---|---|
| N | 12 (4 males) | 28 (11 males) | 40 (15 males) |
| Age | 32.28 (4.81) | 21.77 (3.33) | 24.92 (6.16) |
| English receptive vocabulary [a] | 90.33 (19.22) | 114.59 (9.92) | 107.13 (17.40) |
| Spanish receptive vocabulary [b] | 92% (5%) | 82% (6%) | 85% (7%) |
| Nonverbal intelligence [c] | 100.25 (13.71) | 103.82 (16.61) | 102.8 (15.7) |
| English self-rated speaking [d] | 7.50 (1.51) | 9.68 (0.48) | 9.03 (1.35) |
| English self-rated understanding [d] | 7.33 (1.87) | 9.71 (0.46) | 9.00 (1.54) |
| Spanish self-rated speaking [d] | 9.08 (1.00) | 7.89 (1.29) | 8.25 (1.32) |
| Spanish self-rated understanding [d] | 9.25 (0.87) | 8.07 (1.36) | 8.43 (1.34) |

[a] Indexed with Peabody Picture Vocabulary Test (PPVT-4) standard score. [b] Indexed with Test de Vocabulario en Imagenes Peabody (TVIP) percent accurate. [c] Indexed with Kaufman Brief Intelligence Test (KBIT-2) Matricies subtest standard score. [d] Self-ratings of language proficiency from the Language Experience and Proficiency Questionnaire (LEAP-Q) on a scale of 0–10.

## 2.2. Stimuli

Fifty-one picturable noun words were selected as stimuli. Half of the chosen words were English-Spanish cognates (e.g., English: *baby*, Spanish: *bebé*) and half were non-cognates (e.g., English: *boy*, Spanish: *niño*). Black and white line drawn images to represent each word were gathered from the International Picture Naming Project database (Bates et al. 2003) and Snodgrass and Vanderwart (1980). Early testing led to some images being less accurately labeled than others. Three participants included in the final dataset responded to all 51 images, but the majority responded to 43 images, after the removal of commonly problematic words (*antelope*, *telephone*, *spark*, *ribbon*, *student*, *sofa*, *hole*, and *party*). One further word, *tower*, was excluded after coding of all participants as it was often inaccurately named. The final analyses included only responses to the selected 42 words (21 cognates and 21 non-cognates).

## 2.3. Procedure

The experimental task was explained to participants in either English or Spanish based on their preference. Participants were told they would see images appear on the screen and that they should construct a sentence using the word for the image they see. They were instructed to not make sentences too simplistic (e.g., "This is a cat."), but rather create a complex sentence that included the image label (e.g., "My cat likes to play with yarn and watch birds through the window."). Participants then performed three practice trials using images that were not included in the test trials.

Following the practice trials, participants performed the sentence production task with the full set of images repeated in three blocks. The first two blocks were counter-balanced between participants, such that half of the participants were asked to produce sentences only in English for the first block, and only in Spanish for the second block, and half of the participants did the reverse. The third block was a mixed block where participants were given instructions that they should use English and Spanish in equal proportion (50/50) in their answers. Participants were not instructed on how to use the two languages, other than to use them equally. All 43 images were presented in a random order in each block. Participant responses were recorded using a microphone sitting on the desk in front of them. The instructions provided during the task are available in the OSF repository (https://osf.io/9r4f2/, accessed on 27 September 2022).

## 2.4. Coding

Recordings were transcribed using CLAN conventions (MacWhinney 2014) by a bilingual research assistant. Languages switches were coded using the @s code to denote a switch from the language of the first word in the sentence. A code-switch was defined as use of both English and Spanish anywhere within the response, regardless of where it happened. Some participants gave more than one sentence answers. In these cases, all words were used in determining code-switching. Nine-percent of the data was transcribed a second

time by a set of 3 transcribers to check inter-rater reliability. Cohen's Kappa indicated substantial agreement for the Spanish ($\kappa = 0.66$, $p < 0.001$) and mixed ($\kappa = 0.68$, $p < 0.001$) condition trials. Cohen's Kappa could not be calculated for the English condition as the randomly selected trials were coded as all having no code-switches by both transcribers. Descriptive statistics showed that on average, in the mixed condition, participants used their dominant language 44% of the time in their responses in the mixed condition and their non-dominant language 56% of the time.

Three types of disfluencies, including revisions/retracing, pauses, and filler words, were also coded within the transcriptions. Revisions or retracing included a sentence being started and then all or part of the sentence being restarted (e.g., "I ate <an orange> [//] a peach yesterday"). Pauses were defined as an interruption in speech that was 400 ms or longer (Derwing et al. 2009). Coders measured pause duration in Audacity (Audacity Team 2019). Finally, filler words or filled pauses were defined as any hesitations which included vocalization or meaningless sound (e.g., 'um', 'uh', 'mm'). Intraclass correlations using two-way random effects and a single-rater unit were computed between the main transcriber and the second transcribers for total disfluencies. Results showed moderate to good agreement for the English ($\kappa = 0.80$, $p < 0.001$), Spanish ($\kappa = 0.85$, $p < 0.001$), and mixed ($\kappa = 0.65$, $p < 0.001$) condition trials. Complete transcription instructions are available in the OSF repository.

Trials were also coded for whether the target word was correctly used in the response. Some alternatives from the main list were accepted, including plural forms. For non-cognate words, alternative labels that were also non-cognate words were accepted as correct. A complete list of accepted and not accepted words are available in the OSF repository. Further, a single bilingual research assistant transcribed all words in non-cognate trials for cognate status. An additional two research assistants coded 10% of the non-cognate trials for reliability. They each had perfect ($\kappa = 1$, $p < 0.001$) or near perfect ($\kappa = 0.95$, $p < 0.001$) agreement with the original coder. Proper names and false friends were not counted as cognates.

*2.5. Analysis*

First, trials where the participant did not correctly label the target word, incorporating the alternatives described above, were excluded. Additionally, for a given target word, if a participant did not accurately label the target word, according to above criterion, in both languages, then all three instances of the word were excluded. From the original dataset, this led to the exclusion of 39% of trials. Finally, any non-cognate trials that included a cognate word produced in any part of the response were excluded. This led to the exclusion of 19% of trials from the already cleaned dataset. The final data-set included 2478 trials. Full analyses were performed in R (v4.0.3; R Core Team 2020) using the afex (v0.28.1; Singmann et al. 2021) and lme4 packages (v1.1.26; Bates et al. 2015).

Two properties of the responses, code-switching and disfluencies, were examined in separate analyses. Code-switching was coded as 1 (a code-switch present) or 0 (no code-switch present). Two variables had to be created to examine the effect of disfluencies because of the large number of trials that had no disfluencies (47% of data in final dataset). The first variable indexed presence of disfluencies and was coded as a 1 (disfluency present) or 0 (no disfluency present). The second variable indexed magnitude of rate of disfluencies for trials where a disfluency was present and was calculated based on the combination of the number of revisions/retraces, pauses, and filler words in a response divided by the total number of words in the sentence to control for the fact that longer sentences would give more opportunities for disfluencies. This control was especially important as on average number of words in responses in the mixed condition ($M = 9.55$, $SD = 5.52$) was higher than that in the dominant ($M = 8.56$, $SD = 4.79$) and nondominant condition ($M = 8.23$, $SD = 4.16$), giving participants more opportunities to produce difluences. The second disfluency variable was log transformed to account for the positive skew of the data. Two independent variables were examined in all analyses. Cognate status was coded as $-0.5$ for non-cognates and 0.5 for cognate trials. Condition had three levels: dominant, non-dominant, and mixed and was sum coded. Condition was coded based on the participant's

self-reported language dominance and the block of the task. For example, for a Spanish-dominant speaker, the Spanish block would be coded as the 'dominant' condition, while the English block would be the 'non-dominant' condition.

A logistic mixed effects model was constructed to test the effects of cognate status, condition, and the interaction between the two on code-switching. A maximal random effect structure including a random by-participant intercept, random by-participant slopes for cognate status and condition, a random by-item intercept, and a random by-item slope for condition was initially modeled. In order to allow for model convergence, the final model only included a random by-participant intercept. Model assumptions were tested and the model was configured using the DHARMa package (v0.4.1; Hartig 2021). Significance was tested through likelihood ratio tests. Tukey tests for follow-up analyses on significant effects were done in the emmeans package (v1.5.4; Lenth 2021). Full model summaries are available in the scripts folder in the OSF repository.

The first model for disfluencies was set up in the same way as the code-switching model, with the same set of final random effects for convergence. The second model for disfluencies was a mixed effects regression model with the same fixed effects as the previous two models. A more complex random effects structure converged, so the final model included a random by-participant intercept, random by-participant slopes for cognate status and condition, and a random by-item intercept. Assumptions were checked in part by using the lattice (v0.20.41; Sarkar 2008), car (v3.0.10; Fox and Weisberg 2019), and HLMdiag (v0.4.0; Loy and Hofmann 2014) packages. Significance was determined with Type III Wald F-tests with Kenward-Roger degrees of freedom. Follow-up analyses were reported in the same way as the previous models.

## 3. Results

### 3.1. Code-Switching

The final model showed a significant effect of condition ($\chi^2(2) = 372.04$, $p < 0.001$), but no significant effect of cognate status ($\chi^2(1) = 0.01$, $p = 0.93$), and no significant interaction between the two ($\chi^2(2) = 0.89$, $p = 0.64$). The effect of condition was such that there were overall significantly more code-switches in the mixed condition ($M = 0.18$, $SD = 0.38$) than the dominant ($M = 0.00$, $SD = 0.07$, $OR = 0.001$, $SE = 0.001$, $z = -8.21$, $p < 0.001$) or non-dominant ($M = 0.01$, $SD = 0.11$, $OR = 0.002$, $SE = 0.001$, $z = -9.34$, $p < 0.001$) conditions. Switches in the dominant and nondominant condition did not significantly differ ($OR = 0.45$, $SE = 0.39$, $z = -0.94$, $p = 0.62$). See Figure 1.

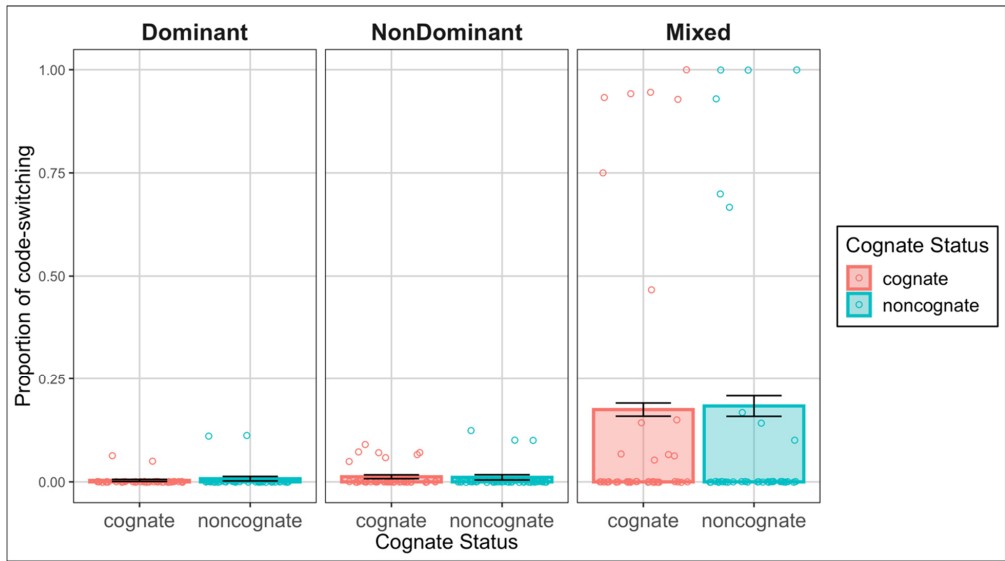

**Figure 1.** Average proportion of code-switching by condition and cognate status with standard error bars. Data points are aggregated by participant.

*3.2. Disfluencies*

The first model for disfluencies revealed a significant effect of condition ($\chi^2(2)$ = 142.19, $p$ < 0.001), but no significant effect of cognate status ($\chi^2(1)$ = 1.34, $p$ = 0.25) and no significant interaction ($\chi^2(2)$ = 3.06, $p$ = 0.22). The effect of condition was such that participants were much less likely to produce a disfluency anywhere in the utterance in the dominant condition ($M$ = 0.38, $SD$ = 0.48) than the nondominant ($M$ = 0.61, $SD$ = 0.49, $OR$ = 0.27, $SE$ = 0.03, $z$ = −10.47, $p$ < 0.001) and mixed conditions ($M$ = 0.59, $SD$ = 0.49, $OR$ = 0.30, $SE$ = 0.04, $z$ = −9.60, $p$ < 0.001). There were no significant differences in probability of producing a disfluency between the non-dominant and mixed conditions ($OR$ = 1.09, $SE$ = 0.13, $z$ = 0.74, $p$ = 0.74).

The second model, examining rate of disfluencies for trials where at least one disfluency was present revealed a significant effect of condition ($F(2, 43.8)$ = 23.70, $p$ < 0.001), no significant effect of cognate status ($F(1, 199.0)$ = 0.86, $p$ = 0.35), and a marginal interaction between the two ($F(2, 1253.0)$ = 2.84, $p$ = 0.06). The significant effect of condition was such that there were more disfluencies in the nondominant language condition ($M$ = 0.32, $SD$ = 0.26) than in the dominant ($M$ = 0.19, $SD$ = 0.14, $b$ = −0.36, $SE$ = 0.06, $t(43.2)$ = −5.71, $p$ < 0.001) and mixed condition ($M$ = 0.25, $SD$ = 0.19, $b$ = 0.24, $SE$ = 0.05, $t(45.7)$ = 5.33, $p$ < 0.001). There was no significant difference between the dominant and mixed conditions ($b$ = −0.12, $SE$ = 0.07, $t(45.1)$ = −1.72, $p$ < 0.001). See Figure 2. The marginal interaction was driven by the fact that there were more disfluencies for cognate than noncognate trials in the mixed condition ($b$ = −0.10, $SE$ = 0.07, $t(121.1)$= −1.57, $p$ = 0.12) but there were more disfluencies for noncognate than cognate trials in the dominant ($b$ = 0.07, $SE$ = 0.08, $t(199.3)$ = 0.89, $p$ = 0.38) and nondominant conditions ($b$ = 0.06, $SE$ = 0.06, $t(93.4)$ = 0.97, $p$ = 0.33). However, the effect of cognate status was not significant for any of the conditions.

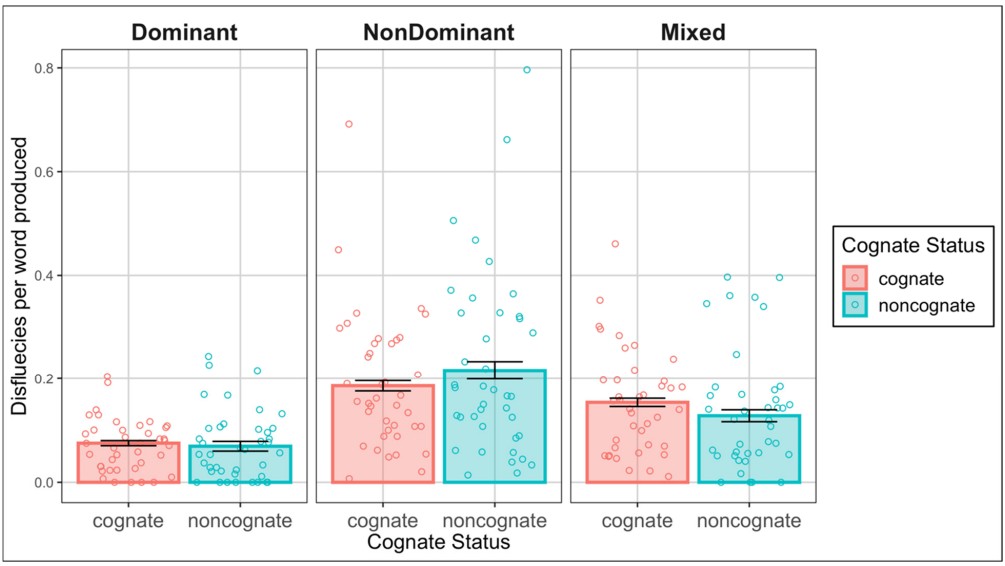

**Figure 2.** Average disfluencies per word by condition and cognate status with standard error bars. Data points are aggregated by participant.

## 4. Discussion

The current study aimed to test the following research question: Do cognates lead to more code-switches and disfluencies in various language contexts in a cued sentence production task? We did not find that cognates led to more code-switches either in participants' dominant or non-dominant language, or when participants were explicitly asked to use both of their languages to a similar extent. Additionally, we did not find that cognates led to more disfluencies, overall. However, there was a trend, albeit a non-significant one, for more disfluencies to occur on cognate trials in the mixed-language condition, and on non-cognate trials in the dominant and the non-dominant language conditions.

Our findings suggest that cognates seem relatively unrelated to code-switching during a cued picture-naming task. Based on effects observed in corpus data, we can make a conjecture that the results of previously reported corpus analyses (Broersma and De Bot 2006; Broersma 2009; Broersma et al. 2020) may indicate that a code-switch leads to using cognates, rather than the opposite. This would support the hypothesis that as a bilingual speaker is planning a code-switch (e.g., Beatty-Martínez et al. 2020; Green and Wei 2014), both languages are more strongly activated, which eases the access to words like cognates, that span the two languages, leading to using them more in speech production as a result.

In our study, bilinguals were equally likely to switch to dominant language when generating sentences in their non-dominant language and to switch to non-dominant language when generating sentences in their dominant language. While such balanced patterns of switching have also been observed in the literature (e.g., Costa and Santesteban 2004; Costa et al. 2006), in our study, the lack of difference is likely due to floor effects in the dominant and the non-dominant language conditions, with very few code-switches produced in both. This is evidence that the participants followed the instructions given to them, as we asked participants to only use one language in the dominant and non-dominant language conditions. Another reason for the lack of significance for this effect and for all interactions across the analyses is that we may not have had the power to detect the effect. Our effect size estimates were based on corpus studies, and while our sample size was sufficient to detect an interaction with a medium effect size, it is possible that the effect size for how cognates affect switching and disfluencies in experimental datasets is much weaker and would require a much larger dataset to detect. One way to improve power would be to raise the number of trials in each condition, although our design included 120 trials per participants, a higher number than average for bilingual adult sentence production studies (e.g., Li and Gollan 2018, 2021). However, to avoid fatigue that is quick to appear in sentence production tasks, more images to be named may require a focus on only the code-switching condition. This could provide a balance between sufficient amount of trials while avoiding a threat to validity resulting from fatigue.

Additionally, in our study, participants did not have an interlocutor, and the presence of an interlocutor in a more ecologically valid interaction task may have yielded effects of cognates on code-switching that the current study did not. While this experimental task allowed to establish a baseline, non-interactive measure of whether cognates lead to a code-switch in production, it should be built upon to incorporate more naturalistic communicative environments. Findings from only one previous study that involved a conversation context showed that cognates could lead to a code-switch, but with the caveat that a switch had to have occurred in the previous sentence used (Kootstra et al. 2020). These findings suggest that some priming and/or alignment occurred in conversation, possibly as a result of audience design strategies. Therefore, future studies would need to try disentangling the roles of cognates, priming and alignment/audience design in a sentence production task. It would also be interesting to combine corpus data and experimental data within participants, to compare cognate effects on code-switching in natural language production with more controlled language production. While corpus studies do not allow establishing causality in the relationship between cognate and code-switching production, the cued production tasks in an experimental setting do not reflect natural communication demands, and may obscure the effects under study (de Bruin et al. 2018; Jevtović et al. 2020). Studying both in parallel in the same participants would provide complementary data to gain a comprehensive overview of code-switching behavior.

One element that we did not examine in this study was inter-sentential code-switching behavior, or spillover effects of the task across trials. We only analyzed code-switches that took place within sentences on an individual trial, which explains why we only observed 15–20% of code-switching in the mixed condition compared to the other conditions. Participants were instructed to use both their languages to an equal extent in this condition and some implemented the instructions within trials, while others alternated language use between trials. We were not able to examine spillover effects because we presented our picture

cues in a random sequence, with cognate and non-cognate trials following each other in a different, non-principled order across participants. An important follow-up study would implement blocked presentation of cognate and non-cognate stimuli, counterbalanced across participants. This would allow observing whether cognates trigger code-switching in a more cognate-dense environment versus a non-cognate environment. Alternatively, pictures cueing cognates versus non-cognates could be more strategically interleaved to gain insights into both intra- and inter-sentential code-switching in the presence of cognates. It is possible that cognates lead to coactivation across multiple sentences, in a further extension of the adjusted triggering hypothesis. Studies have not yet experimentally investigated the extent to which trigger words can influence the presence of code-switches after *n* number of sentences. However, research on the duration of cross-language priming effects suggest that priming is strongest immediately following a prime, and while the priming effect can last, its strength tends to decay rapidly, in a matter of seconds (e.g., Bernolet et al. 2016).

Turning to the covert effects cognates might have on production, we found that disfluencies were highly related to word-finding difficulties, because we observed more disfluencies in the least dominant language, and participants were less likely to be disfluent in their dominant language. This is unsurprising since participants' vocabulary scores and self-ratings of language proficiency were generally lower in the non-dominant language than the dominant language, and lower fluency is one facet of proficiency. We did observe a non-significant interaction effect between cognate status and condition, such that there was a tendency for bilinguals to be more disfluent on cognate than non-cognate trials in the mixed-language condition, and on non-cognate than cognate trials in the single-language conditions. It is possible that the effect of both the mixed-language condition and of cognates compounded, setting a dual-language context where the bilinguals' two languages were highly active and ready to be used. This in turn could have interfered with bilinguals' ability to exert control over their productions, yielding more disfluencies. By contrast, in the dominant- and the non-dominant conditions, without the additional pressure to switch, the production of cognates may have eased production (in line with, e.g., Costa et al. 2000; Li and Gollan 2021), facilitating fluency. This pattern of findings must be interpreted cautiously, given its marginal nature. Nevertheless, the finding indicates that cognates do not always facilitate production (see also Acheson et al. 2012; Broersma et al. 2016), and that language environment and its demands on language control contribute to cognate effects.

In this study, we did not examine whether disfluencies were related to code-switching because we hypothesized that a cognate could lead to more subtle, covert, language production effects, such as disfluencies, instead of leading to overt code-switching. Previous research has found that disfluencies, in the form of hesitations and monitoring behavior (e.g., false starts), co-occur above chance with code-switches (Hlavac 2011). In future research, it would be worth examining this potential relationship between code-switching and disfluencies in relation to language dominance, and the extent to which the presence of cognates or trigger words might moderate a link between the production of disfluencies and code-switches. To further the work on bilingual fluency and its susceptibility to cognate effects, Clyne's corpus data (Clyne 2000), which was not quantitative, could be statistically analyzed to investigate whether the observation that more disfluencies occur in the vicinity of trigger words would hold when examined quantitatively. Moreover, analyzing corpus data which includes transcription of disfluencies could provide information as to the circumstances or speaker characteristics that lead to more or less disfluencies in discourse.

A potential limitation of this study is that global priming effects may have been at play as the study design was within-subjects. We mitigated the impact of these potential effects by having half of the participants complete the task with the English condition first, and half with the Spanish condition first, and all participants finished with the mixed condition. However, we acknowledge that instructing participants on what language to use still does not necessarily prevent subconscious priming effects (e.g., Shin 2008). However, a between-subjects design with bilinguals tends to introduce multiple sources of variability,

which limits the interpretability of findings, due to fluctuations in proficiency, dominance, and other language and social experiences that wary widely across bilinguals and that are very likely to influence switching behaviors. Nevertheless, future studies should attempt between-subjects manipulations with random assignment in order to avoid global priming effects that may have influenced the results in the current study.

## 5. Conclusions

In conclusion, our findings revealed that cognates overall did not trigger more code-switches or disfluencies, regardless of participants' language dominance, in a within-subjects experimental design. However, cognates in a dual-language context tended to lead to more disfluencies in our study. More research combining corpora and experimental design is needed to comprehensively examine how cross-language overlap effects interact with the production of switches and disfluencies across bilinguals' two languages.

**Author Contributions:** Conceptualization, M.M. and M.K.; Data curation, M.M. and A.N.; Formal analysis, M.M.; Visualization, M.M.; Project administration, M.K.; Supervision, M.K.; Writing—original draft, A.N. and M.M.; Writing—review & editing, A.N., M.M. and M.K.; Funding acquisition, M.K. All authors have read and agreed to the published version of the manuscript.

**Funding:** This research was funded by the NIDCD, grant numbers R01 DC016015 and R01 DC011750, and by a Waisman Core Grant from the NICHD, number P30 HD003352.

**Institutional Review Board Statement:** The study was conducted in accordance with the Declaration of Helsinki and approved by the Institutional Review Board of the University of Wisconsin–Madison ("Effects of Bilingualism on Learning and Memory", 2013-0984).

**Informed Consent Statement:** Informed consent was obtained from all subjects involved in the study.

**Data Availability Statement:** Methods, Data, Scripts and Figures are available at: https://osf.io/9r4f2/, accessed on 27 September 2022.

**Acknowledgments:** We acknowledge McKenzie Klein for her help with development of the task, testing participants, and transcribing recordings.

**Conflicts of Interest:** The authors declare no conflict of interest.

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
