# Peer review of "Testing the Triggering Hypothesis: Effect of Cognate Status on Code-Switching and Disfluencies"

_languages, doi:10.3390/languages7040264_

Round 1

Reviewer 1 Report

The authors test whether cognate words trigger code-switching or disfluencies in English-Spanish bilinguals using picture-naming task in three conditions (English, Spanish, English+Spanish). Interestingly, the results do not show clear correlation with cognate words (which may be because of low ecological validity of the experiment), as opposed to previous research, using corpus data. However, more disfluencies occured with cognates in "English-Spanish" condition.

General comments:

1. The author(s) discuss that corpus data cannot show causal connections between a cognate word and a switch (p2 lines 65 ff.) . I think this should be discussed in more detail, and the footnote 1 should be inserted in the main text. Maybe the author(s) could elaborate more on how producing trigger words in the experiments differs from the corpus data and why here the causal connection will be evident. They may also discuss the ecological validity here as well.

2. Some information is presented twice. It is very strange to see "current study" in the plain text (line 140), especially when it was already described. Lines 141-143 repeat the information above. Lines 153-156 also presents the same thing twice (cf. lines 136-139).

3. The author(s) should mention the type of their experiment in the conclusion (p10 lines 473-474), otherwise the results seem too broad. They may also add the indication that this experiment was not ecologically valid.

Specific comments:

p1 lines 6-7: it would be better not to repeat "corpus analyses" twice

p1 lines 25-26: the sentence "where a bilinguals’ two language are simultaneously active" does not seem to be grammatically correct

p1 line 39: the author(s) should indicate which languages were studied by Broersma and De Bot (2006)

p 1 line 44: "the results"?

p2 line 92: maybe the author(s) should explain in parentheses what "shadowing task" means. 

p3 line 100: is the word "confederate" the most suitable in this context?

p3 line 102: is the use of "vs." correct in this context? maybe the author(s) should use "or"? if not, then what about the "control" which was mentioned before?

p3 line 108: "warrants"?

p4 line 171: is "a" needed in this context?

p4 lines 178-179: the repetition of "included" (and this word was also used a lot in the previous paragraph)

p4 lines 190-191: the words "baby", "bebé", "boy", "niño" would look better in italics (as well as the words below, lines 196-197)

p9 line 394, p10 line 444: a strange long space before the words "However" and "We", respectively.

p10 line 467: the author(s) should again indicate the year of the paper by Clyne

Author Response

General comments:

  1. The author(s) discuss that corpus data cannot show causal connections between a cognate word and a switch (p. 2 lines 65 ff.) . I think this should be discussed in more detail, and the footnote 1 should be inserted in the main text.
  • We thank the Reviewer for this comment, and added the footnote in the body of the text.

 Maybe the author(s) could elaborate more on how producing trigger words in the experiments differs from the corpus data and why here the causal connection will be evident. They may also discuss the ecological validity here as well.

  • We added two sentences (l. 151-55) to elaborate on this point and discuss ecological validity: “The present study was designed to examine both whether cognates would trigger code-switching in a cued sentence production paradigm, and whether cognates would lead to an increase in the amount of disfluencies. In the cued sentence production paradigm, bilingual participants were cued with a picture that represented either a cognate or a non-cognate word. This cueing paradigm offers one advantage over examining the natural occurrence of code-switching in corpus data in that we have control over the directionality of the effect (with the limitation of potential spill-over/priming effects between trials). However, the disadvantage of such an approach is that increasing control over the conditions of pro-duction decreases the ecological validity of the task.”
  1. Some information is presented twice. It is very strange to see "current study" in the plain text (line 140), especially when it was already described. Lines 141-143 repeat the information above. Lines 153-156 also presents the same thing twice (cf. lines 136-139).
  • We removed the section title “Current Study” and the repetition l. 136-9 and l. 141-3.
  1. The author(s) should mention the type of their experiment in the conclusion (p10 lines 473-474), otherwise the results seem too broad. They may also add the indication that this experiment was not ecologically valid.
  • We added the experiment type in the first sentence of the conclusion: “In conclusion, our findings revealed that cognates overall did not trigger more code-switches or disfluencies, regardless of participants’ language dominance, in a within-subjects experimental design.”
  • We discussed the pull between control and ecological validity in l. 151-55 in the introduction.

Specific comments:

  • We have addressed each of the comments below – we thank the Reviewer for pointing them out.

p1 lines 6-7: it would be better not to repeat "corpus analyses" twice

p1 lines 25-26: the sentence "where a bilinguals’ two language are simultaneously active" does not seem to be grammatically correct

p1 line 39: the author(s) should indicate which languages were studied by Broersma and De Bot (2006)

  • We additionally commented on the implications of that language pair: “…in a Dutch-Moroccan Arabic corpus. (l.41).

p 1 line 44: "the results"?

p2 line 92: maybe the author(s) should explain in parentheses what "shadowing task" means. 

  • We have explained the shadowing task in the next sentence: “Bilinguals were presented with sentences in their L1 and L2, and asked to listen to recorded speech and to subsequently repeat it as fast and accurately as possible.”

p3 line 100: is the word "confederate" the most suitable in this context?

  • We have reused the word used in Kootstra et al. (2020), but we are open to other suggestions.

p3 line 102: is the use of "vs." correct in this context? maybe the author(s) should use "or"? if not, then what about the "control" which was mentioned before?

p3 line 108: "warrants"?

p4 line 171: is "a" needed in this context?

p4 lines 178-179: the repetition of "included" (and this word was also used a lot in the previous paragraph)

p4 lines 190-191: the words "baby", "bebé", "boy", "niño" would look better in italics (as well as the words below, lines 196-197)

p9 line 394, p10 line 444: a strange long space before the words "However" and "We", respectively.

p10 line 467: the author(s) should again indicate the year of the paper by Clyne

Reviewer 2 Report

This manuscript discusses a study with 40 Spanish-English bilinguals describing pictures in their dominant language, non-dominant language, or in both languages. It specifically assesses the role of cognates, which have been linked to code switches in the triggered switching theory. Code switches as well as disfluencies are analysed. Participants were more likely to switch languages in the mixed-language condition (as expected given the instructions). They were also less fluent in the non-dominant language condition. No significant effects of cognates were found.

This is a really interesting and well written manuscript. Analysing both switches and disfluencies in one paradigm is a great approach as it could potential reveal more subtle effects of cognates and/or mixed-language contexts. Using a picture-naming paradigm allowed for a certain degree of experimental control (over cognate words) while still eliciting relatively free production.

While there is a lot to like, I also have some major comments. Some ask for clarification; others (regarding the number of trials and interpretation) are more substantial. 

Introduction

  • Page 2 discusses evidence from experimental work looking at cognates. These seem to use very different measures (e.g., switching moment or switching costs), which in turn can explain discrepancies across studies. It would be good to be clearer about which types of measures are assessed in the studies reviewed. 

Methods:

  • participants were classified as English or Spanish dominant based on self report (line 175). Was this just based on one question asking them to say which language they are more dominant in? Did the classification align with proficiency (self rated and as assessed in the vocabulary tests)?
  • Procedure: if I’m not mistaken each language condition had 42 trials with each picture presented once in each block. It would be good to state this more explicitly though.
  • This is one of my larger comments: I’m a little worried about power based on the number of trials. 40 participants is a good amount, but the number of trials seems quite low. Line 259 says 2478 trials were left after preprocessing. With 40 participants, 3 language blocks, and two levels of cognate status, this seems to leave around 10 cognate and 10 non-cognates per language condition per participant. This is assuming an equal split between cognates and non-cognates; was that indeed the case after preprocessing? I’m worried that this is a very low number of trials, leaving the study potentially quite underpowered. For example, Brysbaert & Stevens (2018) recommend 1600 observations per condition, which would require 40 trials per condition in this case. A thorough discussion of how sample size/number of trials was determined and potential issues with it being underpowered is needed.
  • If I understand the task correctly, participants had quite a lot of freedom in how they described the images. I might have missed this, but were the type of responses given (e.g., number of sentences and words, complexity of the sentences) comparable across the three conditions? It would be good to at least see the mean number of words generated per trial for the three conditions. If possible, it would also help to report how often each language was used in the mixed context. 
  • At what point in the sentence did participants typically produce the picture name (i.e., the cognate or non-cognate)? I expect the picture name to typically be produced at the beginning but if there’s any variability, it might be worth examining this further (considering the discussion at the start around e.g., Broersma and de Bot’s findings around trigger words preceding or following switches).
  • Non-cognate sentences that included a cognate were excluded. What happened with cognate trials that included multiple cognates? Did this occur? Were all cognate trials coded the same, regardless of the number of cognates?

Results:

  • last paragraph (from line 329) reports the direction of cognate status in the mixed and single-language conditions. The first sentence says this was not significant but please report the actual stats for these effects discussed here (see also my next points).

Discussion/interpretation:

  • the interaction between cognate status and condition in the second disfluency model seems to be over interpreted in quite a few places. The interaction itself is not significant and the follow-up analyses looking at the effects of cognate status in each condition separately do not seem to be significant either. Based on this, I don’t think anything can be said about cognate status playing a role (although power doesn’t help here…). Several places in the manuscript suggest effects of cognate status though. The following parts would need to be revised to make it very clear no cognate effects were significant: penultimate sentence of the abstract; lines 374-376 (first paragraph discussion); lines 444-457 (discussion - it mentions it is marginal but is otherwise formulated quite strongly).
  • It’s not really a surprise that participants switched more in the mixed-language condition than in the single-language blocks given that they were instructed to only use one language (and thus not to switch) in the single language blocks. It would be good to clarify this in the abstract (line 11-12). In the discussion (line 385-400), the finding that bilingual switching was similar for the dominant and non-dominant language is suggested to be at odds with the literature. However, the current study didn’t really allow for a comparison in switching frequency between the dominant and non-dominant language as participants were basically told not to switch in those contexts (and indeed they didn’t). This just strikes me as participants following instructions rather than an actual comparison that can be made. I would suggest revising this paragraph to make it very clear that people, in line with the instructions, basically didn’t switch and that comparisons with the literature therefore cannot be made. I don’t know if it’s possible, but looking at the language in the mixed-language condition might allow you to compare switches to/from the dominant language versus the non-dominant language. The disfluencies are also more interesting in this respect when comparing the two single-language conditions, although here there might be a confound between fluency and proficiency. 
  • Paragraph lines 385-400: as explained above, this section needs substantial revising but another point to consider is that the studies you describe here use very different measures. The Heredia study looks at naming times (asymmetrical switch cost), which is then compared with switching frequency/moment (Bultena). I don’t think these can be compared this directly in the way done here given the different types of measures (and directions).

Author Response

Introduction

  • Page 2 discusses evidence from experimental work looking at cognates. These seem to use very different measures (e.g., switching moment or switching costs), which in turn can explain discrepancies across studies. It would be good to be clearer about which types of measures are assessed in the studies reviewed. 

    • We thank the Reviewer for this comment and have acknowledged the differences in methods and reviewing these studies, lines 100-2 and 110-111:
      “In contrast, Bultena et al. (2015a) did not observe a cognate effect in their study, which used switching costs as an index of triggering instead of switch placement (Kootstra et al., 2012). Bultena et al. (2015a) examined whether switch costs are affected by language coactivation, using cognates preceding a code-switch in a shadowing task. Bilinguals were presented with sentences in their L1 and L2, and asked to listen to recorded speech and to subsequently repeat it as fast and accurately as possible. Switch costs in both directions were not influenced by the presence of cognates in this study. However, the difference in findings across Kootstra et al. (2012) and Bultena et al. (2015a) could be due to the different methods used to measure switching.
      More recently, Kootstra et al. (2020) tested the presence and frequency…”

Methods:

  • participants were classified as English or Spanish dominant based on self report (line 175). Was this just based on one question asking them to say which language they are more dominant in? Did the classification align with proficiency (self rated and as assessed in the vocabulary tests)?
    • Dominance was classified by participant report from the first question of the LEAP-Q. We have added this detail to the manuscript (p. 4, line 196). The question asks participants to list their languages in order of dominance. The first language listed was considered their dominant language. As can be seen from Table 1 in the manuscript, self-ratings of dominance aligned well with both receptive vocabulary and self-rating speaking and understanding cores. Spanish-dominant bilinguals had higher Spanish receptive vocabulary scores and self-rated scores than the English group (t(24.1) = 4.87, p < .001) and vise-versa (t(13.7) = 4.13, p < .01). These statistics have been added to the text to corroborate that language dominance and proficiency align (p. 4, from line 208).
  • Procedure: if I’m not mistaken each language condition had 42 trials with each picture presented once in each block. It would be good to state this more explicitly though.
    • We have changed the wording in the procedure to make this clearer (line 263).
  • This is one of my larger comments: I’m a little worried about power based on the number of trials. 40 participants is a good amount, but the number of trials seems quite low. Line 259 says 2478 trials were left after preprocessing. With 40 participants, 3 language blocks, and two levels of cognate status, this seems to leave around 10 cognate and 10 non-cognates per language condition per participant. This is assuming an equal split between cognates and non-cognates; was that indeed the case after preprocessing? I’m worried that this is a very low number of trials, leaving the study potentially quite underpowered. For example, Brysbaert & Stevens (2018) recommend 1600 observations per condition, which would require 40 trials per condition in this case. A thorough discussion of how sample size/number of trials was determined and potential issues with it being underpowered is needed.
    • We appreciate the reviewer’s valid concerns about power. We did set up the experiment to have more power. With our specific mixed model and set-up, we had the power to detect a small-medium effect size of the interaction of interest (d=.35) according to the PANGEA app (Westfall, 2016). However, we did not foresee quite how much data would be excluded. With our current sample size, we would only be able to detect a medium effect size (d=.47).
    • Due to the fact that we excluded more noncognate trials than cognate trials, there were more cognate trials available in the dataset for all conditions (total cognate trials: 1701, non-cognate trials: 772). For individual participants, the number of trials in the final dataset varied between 1-20 with fairly equal distribution across the whole range. However, mixed effect models are robust to such differences in sample size across conditions, so the overall sample size is likely a larger worry than the exact distribution of trials across conditions. We have added a discussion of how the sample size could affect interpretation of the results (line 484 ff.).
  • If I understand the task correctly, participants had quite a lot of freedom in how they described the images. I might have missed this, but were the type of responses given (e.g., number of sentences and words, complexity of the sentences) comparable across the three conditions? It would be good to at least see the mean number of words generated per trial for the three conditions. If possible, it would also help to report how often each language was used in the mixed context. 
    • The reviewer is correct that we wanted to give participants freedom to produce sentences as would be most natural to them as long as they produced the target word. We can provide more information about the sentences produced in each condition. We created a table to compare some of the suggested trial-level characteristics in the final dataset. We added the percent of each language used in the mixed condition to the manuscript (lines 279-281). We added the average number of words in each condition and the percentage of the dominant languages used in the mixed condition to the manuscript as well (lines 332-335). The issue of more words in utterances in the mixed condition is one of the reasons we used a disfluencies/total words measure rather than a raw disfluency measure.

Dominant

Non-dominant

Mixed

Number of sentences

1.05 (0.22)

1.06 (0.25)

1.08 (0.29)

Number of words

8.56 (4.79)

8.23 (4.16)

9.55 (5.52)

% Dominant language used

0.995 (.06)

0.01 (0.08)

0.44 (0.46)

  • At what point in the sentence did participants typically produce the picture name (i.e., the cognate or non-cognate)? I expect the picture name to typically be produced at the beginning but if there’s any variability, it might be worth examining this further (considering the discussion at the start around e.g., Broersma and de Bot’s findings around trigger words preceding or following switches).
    • Some participants produced the target word early in the sentence, but many also produced it as the last word in the sentence. This is why on average it was produced between 40-50% into the sentence (see table below).
    • When specifically examining the location of the target word in comparison to when the switch happened, it was determined that it was generally much more common for the switch to happen after the target word (80% switch after), but the likelihood that the switch would happen after the target word did not differ between cognate (M = .80) and non-cognate (M = .77) trials (p = .62). However, due to the small number of trials, this may not be the best data-set to examine this.

Dominant

Nondominant

Mixed

Location of target word (in number of words)

3.59 (2.54)

3.59 (2.53)

3.42 (2.53)

Location of target word (in % into the sentence)

46% (28)

48% (27)

42% (27)

  • Non-cognate sentences that included a cognate were excluded. What happened with cognate trials that included multiple cognates? Did this occur? Were all cognate trials coded the same, regardless of the number of cognates?
    • We did consider if we needed to remove cognate trials where other cognates were produced, but determined that it was not necessary. Because participants were given freedom to produce sentences as they like, many sentences had a mixture of cognate and non-cognate words. Since the production of cognates on non-cognate trail was detrimental to our manipulation, we removed them. However, the production of cognates on cognate trials was in-line with our manipulations, therefore in order to not further reduce our sample size, we retained these trials.

Results:

  • last paragraph (from line 329) reports the direction of cognate status in the mixed and single-language conditions. The first sentence says this was not significant but please report the actual stats for these effects discussed here (see also my next points).
    • We edited this last paragraph to include the follow-up statistics for the marginal interaction (lines 392-96).

Discussion/interpretation:

  • the interaction between cognate status and condition in the second disfluency model seems to be over interpreted in quite a few places. The interaction itself is not significant and the follow-up analyses looking at the effects of cognate status in each condition separately do not seem to be significant either. Based on this, I don’t think anything can be said about cognate status playing a role (although power doesn’t help here…). Several places in the manuscript suggest effects of cognate status though. The following parts would need to be revised to make it very clear no cognate effects were significant: penultimate sentence of the abstract; lines 374-376 (first paragraph discussion); lines 444-457 (discussion - it mentions it is marginal but is otherwise formulated quite strongly).

    • We understand the Reviewer’s concerns and have amended our formulation of the results as suggested:
      • In the abstract: “Participants tended to be more fluent on cognate trials in the dominant and the non-dominant condition, and on non-cognate trials in the mixed-language condition, although these effects were not significant.”
      • 1st paragraph discussion: “However, there was a trend for more disfluencies to occur on cognate trials in the mixed-language condition, and on non-cognate trials in the dominant and the non-dominant language conditions, but this effect was not significant.”
      • Discussion: “We did observe a non-significant interaction effect…”

    • It’s not really a surprise that participants switched more in the mixed-language condition than in the single-language blocks given that they were instructed to only use one language (and thus not to switch) in the single language blocks. It would be good to clarify this in the abstract (line 11-12). In the discussion (line 385-400), the finding that bilingual switching was similar for the dominant and non-dominant language is suggested to be at odds with the literature. However, the current study didn’t really allow for a comparison in switching frequency between the dominant and non-dominant language as participants were basically told not to switch in those contexts (and indeed they didn’t).

      • We thank the reviewer for these comments. We have made the clarification in the abstract line: “Unsurprisingly, participants were more likely to code-switch when asked to use both their languages compared to only their dominant or non-dominant language.”
      • We added a sentence at the end of the paragraph in the discussion (l.484-86): “This is evidence that the participants followed the instructions given to them, as we asked participants to only use one language in the dominant and non-dominant language conditions..”

  • This just strikes me as participants following instructions rather than an actual comparison that can be made. I would suggest revising this paragraph to make it very clear that people, in line with the instructions, basically didn’t switch and that comparisons with the literature therefore cannot be made. I don’t know if it’s possible, but looking at the language in the mixed-language condition might allow you to compare switches to/from the dominant language versus the non-dominant language. The disfluencies are also more interesting in this respect when comparing the two single-language conditions, although here there might be a confound between fluency and proficiency. 
    • The direction of switching on mixed trials where there was a code switch was fairly evenly divided between a switch into the dominant language (48% of trials) and a switch into the non-dominant language (52% of trials). This was the case for both cognate (into dominant: 47%) and noncognate trials (into dominant: 50%). Running a model on just this subset of data is difficult due to low sample size but the current trend suggests that there is no difference based on the direction of switch.
    • With respect to disfluencies in the single language conditions, we agree that proficiency is likely the reason for this difference. Indeed, the non-dominant language is the less proficient language for participants as confirmed by their vocabulary scores and self-ratings. This gives us a good baseline to assess if the rate of disfluencies in the mixed condition are high or low, and unsurprisingly, they fall in between the rate of disfluencies in the dominant and nondominant language conditions. We added a sentence in the discussion to highlight this logic (from line 562).
  • Paragraph lines 385-400: as explained above, this section needs substantial revising but another point to consider is that the studies you describe here use very different measures. The Heredia study looks at naming times (asymmetrical switch cost), which is then compared with switching frequency/moment (Bultena). I don’t think these can be compared this directly in the way done here given the different types of measures (and directions).
    • We thank the Reviewer for this comment. We have reviewed the paragraph and removed references to the Heredia and Bultena studies which use different measures than our study.

Reviewer 3 Report

Review

This is good work on disfluency and is very interesting in that respect.  The application of statistics is measured and thorough, and the fact that the materials and data are available online is always indicative of responsible participation in the field.

However, this manuscript has some conceptual issues, related to design, studying bilingualism, and relationship with previous literature.

1. Priming

The largest conceptual issue relates to priming.  Because this study is attempting to address the Adjusted Triggering Hypothesis, priming is central to the validity of this study.

Since language choice is heavily affected by priming, it is very curious that a within-subjects design was chosen.  How can the author(s) be sure that global priming affects weren’t at play?  How did the authors prime the participants into the various language modes (i.e., relative levels of activation)?  They themselves reference this issue within the mixed language block (lines 420-440), but this issue applies throughout.

The critique of priming within the mixed language block can even go further:  rather than priming the participants into activating both languages for the mixed-language block, the authors instruct participants to use both languages equally.  This is an extremely unnatural task, and it is unknown whether participants’ activation of both languages increased over the course of the block.

In general, effects of having to switch languages could be expected going from the dominant to non-dominant blocks as well (c.f. the work of Kroll).

2. Design of task

The cued sentence production task from this study is different from the typical design.  In Koostra et al. (2010), for example, there’s a lead-in fragment, which is a typical way to structure cued sentence production.  The author(s) do not provide a rationale for doing it differently in this manuscript, and the results of the change are unknown.

3. Bilingualism as a construct

One of the challenges of bilingualism as a field is that it receives an array of definitions.  In this manuscript and a subset of the work it references, bilinguals include second language (L2) learners/speakers.  Clyne’s original work (and that of Poplack who gets cited) is about early bilinguals.  Note that Poplack (1980) pointed out that only balanced bilinguals switch in a way that is not constrained by lack of proficiency, meaning that “natural” switches are the product of early, balanced bilinguals.  The application of Clyne and Poplack’s literature to this version of code-switching is apples to oranges.  There is a version of code-switching that happens among L2s, but it is related to disfluency in almost all cases.  That is one of the reasons that disfluency is an important contribution of this study.

Note that the use of self-report for L2 proficiency is problematic.  A secondary measure would have been preferable.

4. Reference to “planning” to code-switch

Lines 381-384:  Other literature does not claim that code-switchers “plan” to code-switch; in fact, this is quite different than the vast body of literature suggesting that priming is at play; clarification needed.

Author Response

  1. Priming

The largest conceptual issue relates to priming.  Because this study is attempting to address the Adjusted Triggering Hypothesis, priming is central to the validity of this study.

Since language choice is heavily affected by priming, it is very curious that a within-subjects design was chosen.  How can the author(s) be sure that global priming affects weren’t at play? 

  • We understand the Reviewer’s concerns as indeed global priming effects could be at play. To counterbalance any potential effect of priming, half of the participants completed the task with the English condition first, and half with the Spanish condition first, and all participants finished with the mixed condition. A between-subjects design with bilinguals is always risky, due to fluctuations in proficiency, dominance, and other language and social experiences that wary widely across bilinguals and that are very likely to influence switching behaviors. We acknowledge this point in the Discussion (p. 11, lines 593-602).

How did the authors prime the participants into the various language modes (i.e., relative levels of activation)?  They themselves reference this issue within the mixed language block (lines 420-440), but this issue applies throughout.

  • Participants were overtly asked to produce in English or Spanish in these conditions, such that participants always knew in what language productions were expected.

The critique of priming within the mixed language block can even go further:  rather than priming the participants into activating both languages for the mixed-language block, the authors instruct participants to use both languages equally.  This is an extremely unnatural task, and it is unknown whether participants’ activation of both languages increased over the course of the block.

  • Though there is no way to directly know the language activation of participants during the experiment, we can examine if the proportion of use of each language across the mixed language block changed over the course of the 42 trials with correlations. The correlation between proportion of dominant language usage and trial number was not significant (r = -.01, p = .75). Similarly, the proportion of Spanish usage and trial number was not significantly correlated (r = .02, p = .26), regardless of whether the participants had heard Spanish on the block before the mixed block (r = .02, p = .56) or if they heard English on the block before the mixed block (r = .02, p = .36). The conclusion we can draw from this is that both languages were likely active during the mixed block just as the instructions would suggest they would need to do. Additionally, activation did not seem to change during the course of the mixed block of the experiment. Any priming from the experiment set up was likely outweighed by the instructions of the task.
  • We have added a sentence and a reference to justify our instructions in the mixed condition lines 163: “The purpose of the instructions in the mixed language conditions was to maximize language switching (Gollan et al., 2014).”

In general, effects of having to switch languages could be expected going from the dominant to non-dominant blocks as well (c.f. the work of Kroll).

  1. Design of task

The cued sentence production task from this study is different from the typical design.  In Koostra et al. (2010), for example, there’s a lead-in fragment, which is a typical way to structure cued sentence production.  The author(s) do not provide a rationale for doing it differently in this manuscript, and the results of the change are unknown.

  • We acknowledge the difference in methods between our study and previous studies. We added an explanation in the introduction l. 145-153: “The present study was designed to examine both whether cognates would trigger code-switching in a cued sentence production paradigm, and whether cognates would lead to an increase in the amount of disfluencies. In the cued sentence production paradigm, bilingual participants were cued with a picture that represents either a cognate or a non-cognate word. This cueing differs from the natural occurrence of code-switching in corpus data in that we have control over the directionality of the effect (with the limitation of potential spill-over/priming effects between trials). In turn, increasing control over the conditions of production lead to a decrease in the ecological validity of the task.
  1. Bilingualism as a construct

One of the challenges of bilingualism as a field is that it receives an array of definitions.  In this manuscript and a subset of the work it references, bilinguals include second language (L2) learners/speakers.  Clyne’s original work (and that of Poplack who gets cited) is about early bilinguals.  Note that Poplack (1980) pointed out that only balanced bilinguals switch in a way that is not constrained by lack of proficiency, meaning that “natural” switches are the product of early, balanced bilinguals.  The application of Clyne and Poplack’s literature to this version of code-switching is apples to oranges.  There is a version of code-switching that happens among L2s, but it is related to disfluency in almost all cases.  That is one of the reasons that disfluency is an important contribution of this study.

  • We added information to paragraph l. 138-141 in the introduction to clarify this:

“The population in Clyne (1980, 2000) and Poplack (1980) are early bilinguals who are less likely to produce disfluencies generally due to high proficiency in both their languages. Disfluencies may be more readily detectable in speakers who are not as proficient in their L2.”

Note that the use of self-report for L2 proficiency is problematic.  A secondary measure would have been preferable.

  • Self-reported proficiency was only relevant for excluding participants. Here we needed a measure that could be comparable across multiple languages (i.e. beyond English and Spanish) in case we needed to exclude participants for having knowledge of another Romance or Germanic language. A self-reported variable was the best way to do this. We should also note that the self-report measure we used has been validated and shows a correlation between self-rated speaking proficiency and performance on standardized tests (Marian, Blumenfeld, & Kaushanskaya, 2007). Additionally, in our sample, when examining the correlation between self-reported speaking abilities in the L2 and the receptive vocabulary results, correlations were around .50. See tables below for all correlations.
  • Self-reported dominance helped determine which language would be a participants’ dominant language. The relation to the self-related dominance grouping and secondary measures—the receptive vocabulary score in Spanish and English—is reported in Table 1 of the manuscript. It is clear that self-reported dominance aligns with both self-reported proficiency scores and the receptive vocabulary scores. Spanish dominant participants have higher Spanish scores and English dominant participants have higher English scores. We can confirm that there was a significant difference in receptive language scores between the Spanish and English dominant bilinguals for both the English (t(13.7) = 4.13, p < .01)) and Spanish (t(24.1) = 4.87, p < .001) tests. This has been added to the text (lines 208-212) Therefore, for the purposes of this study (where the specific proficiency values are not used in any analyses) self-reported dominance was sufficient.

L2 Spanish group

TVIP (Spanish receptive vocabulary proportion correct)

Self-reported Spanish speaking

.47*

Self-reported Spanish understanding

.57**

Self-reported Spanish reading

.27

*<.05, **<.01

L2 English group

PPVT (English receptive vocabulary standard score)

Self-reported English speaking

.49

Self-reported English understanding

.77**

Self-reported English reading

.09

**<.01

  1. Reference to “planning” to code-switch

Lines 381-384:  Other literature does not claim that code-switchers “plan” to code-switch; in fact, this is quite different than the vast body of literature suggesting that priming is at play; clarification needed.

  • We added two references to support this statement l. 475: Beatty-Martinez et al., 2020; Green & Wei, 2014.

Round 2

Reviewer 2 Report

I would like to thank the authors for their revisions. My previous comments have all been addressed sufficiently apart from one. While the authors have added some brief discussion of the study potentially being underpowered, the main issue (as described in my previous review) is the low number of trials per condition. This is still not addressed in the manuscript (or at least, I couldn’t find a thorough discussion).

Author Response

  • We thank the reviewer for encouraging us to be more explicit about power. The production task that we used focuses on accuracy rates rather than RTs. The minimum trial number indicated by Brysbaert & Stevens (2018) was for time-based tasks, accounting for the fact that RT data tend to be very noisy. Our design included 120 trials, and this number is on the high end compared to other bilingual adult sentence production studies (i.e. Li & Gollan, 2018; 2021). We agree that an increase in the number of trials would improve our power and now state so in the Discussion (l. 491-2). However, we also point out that our design compares favorably with prior studies of sentence production (l. 493-4).  
  • We added a few sentences to discuss this specific issue in the same paragraph in which we discussed power in the discussion (l. 484-97): “One way to improve power would be to raise the number of trials in each condition, although our design included 120 trials per participants, a higher number than average for bilingual adult sentence production studies (e.g., Li & Gollan, 2018; 2021). However, to avoid fatigue that is quick to appear in sentence production tasks, more images to be named may require a focus on only the code-switching condition. This could provide a balance between sufficient amount of trials while avoiding a threat to validity resulting from fatigue.”

Reviewer 3 Report

Thanks for responding to my concerns.  The correlations with respect to the items is an important piece of evidence, and I thank the authors for including it.

As a point of clarification, priming is subconscious, and telling the participants what language they should be using is not the same as preventing it. See, for example, Shin's (2008) dissertation.

Shin, J. A. (2008). Structural priming in bilingual language processing and second language learning. University of Illinois at Urbana-Champaign.

Author Response

  • We revised the last paragraph of the discussion and added a sentence to address this comment: “However, we acknowledge that instructing participants on what language to use does not necessarily prevent subconscious priming effects (e.g. Shin, 2008).” (l. 604-5).